# Physiotherapy Students’ Experiences about Ethical Situations Encountered in Clinical Practices

**DOI:** 10.3390/ijerph18168489

**Published:** 2021-08-11

**Authors:** Marta Aguilar-Rodríguez, Kati Kulju, David Hernández-Guillén, María Isabel Mármol-López, Felipe Querol-Giner, Elena Marques-Sule

**Affiliations:** 1Department of Physiotherapy, University of Valencia, Gascó Oliag 5, 46010 Valencia, Spain; marta.aguilar@uv.es (M.A.-R.); felipe.querol-giner@uv.es (F.Q.-G.); 2Faculty of Health and Well-Being, Turku University of Applied Sciences, Joukahaisenkatu 3-5, 20520 Turku, Finland; kati.kulju@turkuamk.fi; 3Nursing School La Fe, Adscript Center of the University of Valencia, Health Research Institute La Fe, Group of Investigation GREIACC, Fernando Abril Martorell 106, Pabellón Docente Torre H, 46026 Valencia, Spain; maribelmrlp@gmail.com; 4Physiotherapy in Motion, Multispeciality Research Group (PTinMOTION), Department of Physiotherapy, University of Valencia, Gascó Oliag 5, 46010 Valencia, Spain; elena.marques@uv.es

**Keywords:** physiotherapy students, ethics, student experiences, clinical practice, qualitative research

## Abstract

(1) Background: It is important to explore the ethical situations that physiotherapy students encountered in their clinical practices. (2) Methods: Qualitative, explorative, descriptive study. The participants included third-year physiotherapy students. They had to write five narratives about ethical situations encountered in their clinical practices. Krippendorff’s method for qualitative content analysis was used to cluster units within the data to identify emergent themes. The study protocol was approved by the authors’ University Ethic Committee of Human Research (H1515588244257). (3) Result: 280 narratives were reported by 64 students (23.34 ± 4.20 years, 59% women). Eight categories were identified from the qualitative analysis of the data: (a) professional responsibility, (b) professional competence, (c), beneficence, (d) equality and justice, (e) autonomy, (f) confidentiality, (g) respect for privacy, and (h) sincerity. All participants were informed and provided written informed consent. (4) Conclusions: Ethical principles were frequently violated in physiotherapy. Experiences of physiotherapy students must be examined to tailor educational interventions prior to their initiation into practice. Ethics education is needed in workplaces and should be increased in basic education. Facilitating the ethical awareness of future physiotherapists is a challenge for university teachers who provide ethical competence training.

## 1. Introduction

Clinical learning by means of clinical practices is an indispensable part of physiotherapy education. Previous studies have observed that the qualities of physiotherapists and the art of providing healthcare are based in the clinical experience and also in the ethics-related background [1,2]. During clinical practices, physiotherapy students experience many different clinical- and ethics-related situations.

An ethical situation (ES) is about a conflict between values. In the discussion of ES, strong emotions and fear of making a bad choice tend to be involved [3]. ES in physiotherapy emerges, e.g., when the patient, often due to scarce resources, does not receive the amount of physiotherapy that is needed [4,5,6,7] or when the therapists prioritize patients based on their socioeconomic status [4,8]. Ethical issues concerning the patients’ self-determination also occur [4,5].

Regarding physiotherapy education, there exists previous research concerning moral reasoning and judgment skills [9,10,11], moral distress [12], and educational interventions over the past ten years [13,14,15,16,17,18]. Nevertheless, studies about ES that physiotherapy students encounter during their clinical practices are scarce [14,19,20]. Geddes et al. (2004) identified three main themes [21]: respect for the uniqueness of individuals, professionalism, and professional collegiality, with minor themes being allocation of resources, advocacy for client, society and/or health policy, and informed consent. According to Larin et al. (2005) [22], inequality, competence, and telling the truth to the patient were the ES’s faced by physiotherapy students during clinical practice.

Physiotherapists are required to make ethical decisions with the support of respective codes of professional ethics, which provide a framework for decision-making in clinical practice. The codes of ethics for physiotherapists in different countries are based on the World Confederation for Physical Therapy ethical principles [23]. Those expect the physiotherapists to respect the rights of individuals; comply with laws and regulations; provide honest, competent and high-quality services; be responsible; and provide accurate information about the services provided [23].

Since physiotherapists encounter a variety of ethical challenges in their practice, ethical competence is needed to respond to those challenges [24]. Physiotherapists consider themselves competent in ethics, even if they are not very familiar with ethical codes or methods for ethical problem-solving [24]. Previous training in ES could help to develop ethical competence and to face these situations in the most suitable way, providing high-quality healthcare [17,18]. 

Experiences of physiotherapy students must be examined to tailor educational interventions prior to their initiation into practice. This study aimed at providing a richer understanding of physiotherapy students’ experiences by exploring the ES’s they encountered in their clinical practices. This study approached the topic from a novel viewpoint, exploring the ethical conduct of the practicing physiotherapists through the eyes of physiotherapy students.

## 2. Methods

### 2.1. Participants

Third-year degree undergraduate physiotherapy students aged 20–30 years were recruited. All enrolled participants provided written informed consent. Inclusion criteria were to be studying for the Physiotherapy Degree at the University of Valencia (Spain) and to have participated in a blended-learning program regarding professional ethics. The students followed clinical practices in hospitals and clinical centers for 8 months. Clinical training of the participants run in rotating periods in various health settings.

### 2.2. Design

An in-depth qualitative exploratory and descriptive study was performed in order to explore ethical issues experienced by physiotherapy students. In this regard, the research was addressed to analyze students’ feelings and perceptions to identify and understand the meanings that are attributed to the focus of the study, i.e., the ethical situations encountered by the students within the clinical setting.

### 2.3. Data Collection

Data were collected via written narratives. Initially, the students were invited to share stories about ES’s, clinical vignettes that illuminated details about ES’s, and examples of ES’s that they witnessed and experienced. Concretely, they were asked to write five narratives about the ES’s they encountered in their clinical practices [25]. Background information such as gender, age, and academic year was also collected.

### 2.4. Data Analysis

Qualitative content analysis was adopted to analyze the written data [26]. The use of content analysis enables the researcher to choose between focusing on manifest (developing categories) and latent content (developing themes). In this study, we focused on manifest content, which is the visible, obvious component of what the text said [27], from which the development of categories was achieved. All narratives were read several times to get an impression of the whole text. When searching for meaning units (words, phrases or paragraphs) that could be connected to the aim, each narrative was seen as the unit of analysis [26]. The identified meaning units were condensed and then coded to reflect the content they represented. In the next step, the condensed meaning was compared across all the written narratives in a search for patterns of similarities and differences. The condensed meanings were then sorted and abstracted into eight categories. Only after the categories were identified and confirmed, quotations were translated into English by the researchers. The direct quotations from the written narratives were extracted to substantiate the categories established. Examples of meaning unit, condensed meaning unit, sub-category and category are shown in Table 1. Several strategies were used to ensure credibility and trustworthiness of the data [28], including multiple research team members reviewing the narratives (MAR, EMS, MIML), multiple team discussions to identify categories, a member checking and coding verification by a second team member (EMS).

### 2.5. Ethical Considerations

The study protocol was approved by the authors’ University Ethic Committee of Human Research (H1515588244257). The teaching methodology was an Educational Innovation Project of the authors’ University (45/FO/18). The study followed the ethical principles according to the Declaration of Helsinki. All participants were informed about the study and procedures, and provided written informed consent. Each student was informed that they could leave the study at any time, and that confidentiality would be maintained during the publication of the study. All students were assured that not participating would not affect their academic standing. In order to ensure the principle of confidentiality, participants’ data were anonymized by using numerical codes and all data were treated confidentially.

### 2.6. Funding

The authors received no financial support for the research, authorship, and/or publication of this article.

## 3. Results

Sixty-four physiotherapy students met the inclusion criteria, completed the study and were included in the analyses (mean age 23.34 ± 4.20 years, 59% women). From the possible 320 initial narratives that should have been reported, 280 were finally written (response rate 87.5%). Eight categories were identified from the analysis of the data: (1) professional responsibility, (2) professional competence, (3) beneficence, (4) equality and justice, (5) autonomy, (6) confidentiality, (7) respect for privacy, and (8) sincerity. Results are described under these categories with quotes to illustrate findings.

### 3.1. Professional Responsibility

A major ES that the majority of participants in this study noted was the lack of professional responsibility of the physiotherapists at the hospitals in which they performed their clinical practices. The most frequently reported situation was that physiotherapists delegated their tasks to the auxiliary staff or even to the patients, considered by the students as an unethical situation. In addition, almost all students mentioned that physiotherapists applied the treatments imposed by the doctor although they did not agree with this decision. Some students described these situations thus:

*“The treatments of patients are established by rehabilitation doctors depending on the pathology, thus there is no individualized treatment. […] The same treatments are applied to each of the patients based on the injury. […] The physiotherapist is aware that the treatment to be applied is not specific and may believe that it is not the most suitable for that patient”* (Participant 9).

*“In the unit of electrotherapy […] the physiotherapists usually explain to their patients how to use an electrotherapy device, specifically the ultrasounds machine, so that the patients apply the treatment to themselves. […] Then, the physiotherapists perform other treatments and therefore there is no accumulation of patients”* (Participant 1).

*“The physiotherapist asked the patients to apply the ultrasounds to themselves”* (Participant 60).

### 3.2. Professional Competence

Almost all students stated that the most frequent ethical situations in their clinical practices were related to the lack of professional competence of the physiotherapists. Many participants of this study denounced that tutors and other physiotherapists often failed to fulfil it, by giving examples such as:

*“A physiotherapist incorrectly placed a pole therapy system to the detriment of the patient’s recovery, creating unnecessary discomfort and pain in the active mobilization of the shoulder, thus applying a non-appropriate treatment”* (Participant 59).

*“A 73-year-old patient who underwent a heart valve surgery […] came to strengthen muscles and recover functional outcomes […] The physiotherapist did not assess the resting heart rate, nor established a safe heart rate range”* (Participant 26).

### 3.3. Beneficence

Students warned that the physiotherapist was not always trying to do the best for each patient (principle of beneficence). Almost all students agreed that physiotherapists should never put their personal interests first, given that their role as healthcare professionals is to provide the best health assistance and thus take care of the patient as best as possible. To this end, the physiotherapists should endeavour to better care for the patients. Several students observed:

*“The number of patients is so high that it is necessary to reduce the treatment time in order to treat every patient. It is company policy to treat many patients in a short time, but we know that longer treatment would be more effective by far”* (Participant 20).

*“The same treatments are applied to each of the patients based on the injury. […] I asked about the possibility of changing treatment, but this was the answer: ‘You should apply the treatment written in the paper, the treatment is not for you”* (Participant 9).

*“A physiotherapist talked about some techniques that would add to a treatment, or even replace the treatment established by the rehabilitation doctor, […] we have observed that they didn’t talk with the doctor to reach an agreement and outline a more appropriate treatment”* (Participant 7).

### 3.4. Equality and Justice

Participants stated that the treatment applied to the patients was not as equitable as they thought it should be, and they often observed situations of favoured treatment and breaching the ethical principle of justice. Some students recognized the challenging position of the physiotherapists who have to do what supervisors ask them to do and therefore may have limited power to make decisions, and also a variety of questionable ethical situations that conveyed discriminatory attitudes toward patients, such as discrimination based on age or race, although these situations were less frequently reported.

*“I have seen privilege treatment towards some patients for different reasons, in this case to the wife of a retired head of the Department of Traumatology […] the duration of the treatment was longer than the rest of the patients with the same pathology, even neglecting other patients due to lack of time”* (Participant 52).

*“Patients are treated depending on the hour they are scheduled at. […] an old woman who was queuing for the microwaves and two people, a doctor and a supervisor of my tutor, entered in the room and sneaked directly to the microwaves […] they did not care about the patients who were queueing for that treatment”* (Participant 15).

*“My tutor treated two amputee patients in a very different way. One suffered from hepatitis C, with alcohol problems caused by alcohol, living in an asylum reception center. The other had lupus erythematosus and diabetes. The treatment of the second patient was much more humane, longer and personalized, while the first one was ordered to do exercises on his own, so the physiotherapist was not involved at all in that treatment”* (Participant 54).

### 3.5. Autonomy

The students noted that it was not a common practice in the physiotherapists’ work environment to ask for patient consent to the assessment or treatment techniques that were to be performed. Some students reported that displaying an empathetic approach should be a key characteristic of the physiotherapists, although sometimes it seems not to be fulfilled. Three students described how a physiotherapist did not respect the autonomy of a patient:

*“We had to apply passive mobilization to a patient and help him walk. […] the patient did not want to collaborate, he was not in the mood […] Although the patient didn’t want to do it anyway and explained this several times, the physiotherapist applied the mobilizations. I think the physiotherapist should have respected the patient’s decision”* (Participant 43).

*“I observed how a physiotherapist performed a treatment although the patient didn’t want to and was not motivated at all”* (Participant 31).

*“Transcutaneous electrical nerve stimulation in the lower back of a patient was applied, thus the lower back had to be uncovered […] The patient presented serious complexes about her physical appearance and preferred another type of therapy in which she did not have to show any part of her body”* (Participant 33).

### 3.6. Confidentiality

Many students experienced several situations in which the physiotherapists told intimate aspects of the patients to other colleagues or to other patients, in breach of the duty of confidentiality. Students recognized the role of the physiotherapists in the application of the healthcare process, but they were clear about the boundaries of the physiotherapist’s role in terms of violating the confidentiality. For instance, two students commented:

*“A physiotherapist was talking to a student while they were treating another patient that had nothing to do with what they were talking about. Professional secrecy was breached, since the patient in front of them did not have to know the characteristics, circumstances or facts of the private life of another person”* (Participant 32).

*“The physiotherapist discussed the clinical case of a patient with another physiotherapist, in front of another patient, without taking care that the new patient should not been listening this information”* (Participant 50).

### 3.7. Respect for Privacy

The hardship of treatment rooms and patients’ rooms was also detailed, in order to respect their privacy. Students focused on patients’ experiences and point of view in this regard and acknowledged the possible consequences and feelings of such incidents. The narratives below illustrate the lack of respect for privacy.

*“A hemiplegic woman […] had to take off her top due to the physiotherapy treatment […] The physiotherapist left the door open thus leaving the patient observed by other patients. The patient’s boyfriend had to solve the problem by giving her a shirt to cover herself and placing a screen in front of the door”* (Participant 22)

*“I have verified how the privacy of some patients has not been respected. This privacy has been violated by the physiotherapist when sharing information with other people […]. In this way, the physiotherapist has violated the principle of autonomy, as well as professional values such as the respect for the person and protection of human rights. Patients have the right to privacy”* (Participant 9).

*“I have seen an elderly man at the hospital room without underwear and with the door open. The family of the patient with whom he shared the room was present, as well as the nurse of that other patient. In this situation, the physiotherapist applied a lower limb mobilization and afterwards asked him to walk with his help. Then, the patient asked to please close the door”* (Participant 59).

### 3.8. Sincerity

The last category, but not the least important because of its cultural and contextual relevant traits, was sincerity. The students observed that some physiotherapists showed a lack of sincerity with their patients, mainly motivated by the ignorance of what type of information they should or should not give them. In this regard, many students considered that lack of sincerity would have far-reaching consequences for the patient, since a non-real prognostic is reported and thus false expectations of cure are given to patients. For example, two students expressed their surprise when experiencing the following situations:

*“In my clinical practices I could see how the physiotherapist lied to a patient who asked her if she was going to recover completely after her fracture of the distal third of the humerus. The physiotherapist stated that she would recover in order not to worry the patient, although afterwards she told me that it was not clear at all”* (Participant 47).

*“A patient with an unusual pathology that had not improved with any treatment in other clinics came to the clinic. The physiotherapist, in a role more of a businessman than a physiotherapist, lied to the patient telling him that he had already treated this condition on several occasions and that he had always achieved the expected results”* (Participant 11).

## 4. Discussion

To our best knowledge, this is the first study exploring the ES’s that physiotherapy students encountered in their clinical practice, focusing on practicing physiotherapists’ conduct. It seems that the most problematic issues concerned professional responsibility (i.e., duty to treat the patient); professional competence (i.e., using the technology and taking account of patient safety); beneficence (i.e., offering the best for the patient); equality and justice (i.e., discrimination due to the patient’s background); autonomy (i.e., issues concerning patients’ self-determination and informed-consent practices); confidentiality in exposing patient matters to an outsider; respect for privacy during the therapy session; and sincerity concerning truth telling to the patient. The findings also indicate that the ethical principles [23] were frequently violated in physiotherapy. This study provided an opportunity to gain a deeper insight into the lived experiences of physiotherapy students, not from a clinical point of view but from an ethical perspective. Additionally, it has been proven that previous training in ethics may lead physiotherapy students to identify ethical situations in their clinical learning practices.

Our findings have some similarities with previous studies pointing out ethical problems that physiotherapists encounter in their practice. Previous research in physiotherapists emphasizes the principle of beneficence in ethical decision-making [29]. It is considered an ideal of being beneficent toward the patient [30]. This study reveals balancing between needs and expectations [31], sometimes acting against the principle of beneficence, being forced to act against their own values. Referring to the findings, physiotherapists should always act in the best interest of a patient, which is a matter of ethical competence to gain the best possible solution for the patient [32]. Equality in patient care or access to physiotherapy was mentioned to be uneven. In addition, a previous study stated that, for example, outside constraints [7] and physiotherapists’ attitudes towards patients due to a patient’s background or illness [4] have an impact in how the therapists prioritize among patients.

Concerning the autonomy of the patient, this study revealed concerns regarding the informed-consent practices. According to the WCPT ethical principles [23], the physiotherapist should provide accurate information about the services provided, and this principle seems to be violated. This finding has similarities with previous studies [22,33,34,35]. Delany et al. (2007) stated that the therapists’ primary concern was to provide information that led to a (therapist-determined) beneficial therapeutic outcome, rather than to enhance autonomous patient choice [33]. Only 41% of physiotherapists are always seeking to obtain informed consent at the onset of physiotherapy [35]. The findings of this study highlight the importance of informed-consent practices in physiotherapy—ethical knowledge and moral reasoning have a positive influence regarding the awareness of informed consent [35].

As this study reveals, the students do recognize ethical problems in their clinical practices and thus may learn from these situations and develop ethical competence from a practical approach. Identifying ethical situations may help students to reflect and develop ethically sustainable care practices in their future daily practice as physiotherapists. Reasons for not reporting ethical misconduct are low position of hierarchy, fear of not being a team player, not recognizing an issue, and personal consequences [36]. These barriers should be pointed out and discussed already during physiotherapy education.

### 4.1. Implications of Findings

This study has far-reaching implications for students and physiotherapists. As shown in the present study, there are ethical issues that may be challenging for physiotherapy students, physiotherapists, and other healthcare professionals. To address these challenges, ethics education cannot be underestimated and should be based on practice-related issues, rather than on rare and controversial theoretical situations. By being aware of the ethical situations that may arise in the clinical environment, students can understand their future ethical role as healthcare providers and take an ethical perspective into account in their future daily clinical practice. Further ethics education is needed in workplaces, and as today’s students are tomorrow’s professionals, more research should explore how these ethical situations impact the patients’ care and treatment adequation.

### 4.2. Strengths and Limitations

A major strength of this study is that this is the first study to explore ethical situations encountered by physiotherapy students in their clinical placement, focusing on practising physiotherapists’ conduct. The number of narratives provided could also be highlighted. As limitations, since the research sample is from a single university in Spain it is difficult to generalize the results to other countries. It is also worth noticing that as the students have observed physiotherapists in their practice, the findings are based on students’ interpretation of a certain situation, according to their current understanding. Ethical issues in clinical environments can be very complicated, and choosing how to act depends on how a situation is interpreted.

Future research about differences among ethical situations perceived by students of different academic years could be necessary to find out if their experience in the clinical setting helps them to be more or less critical from an ethical point of view. Quantitative research about the association existing between ethical situations encountered by physiotherapy students and level of knowledge in ethics could be implemented. Future studies could use mixed methods, comparing results for quantitative and qualitative content.

## 5. Conclusions

Our findings indicated that ethical principles were frequently violated in physiotherapy. Eight categories were identified from the analysis of the data: professional responsibility; professional competence; beneficence; equality and justice; autonomy; confidentiality; respect for privacy; and sincerity. Additionally, this study provided insights into physiotherapy students’ experiences when identifying ethical situations in their clinical learning practices, thus experiences of physiotherapy students in this regard should be examined to tailor educational interventions prior to their initiation into practice. Facilitating the ethical awareness of future physiotherapists is a challenge for university teachers who provide ethical competence training. Future research should explore how these ethical situations impact on the patients’ care and treatment adequation.

## Figures and Tables

**Table 1 ijerph-18-08489-t001:** Example of forming categories from content analysis of narratives about ethical situations encountered by physiotherapy students.

Step	Analyses Process	Example
1	Meaning unit	“The physiotherapist took a medical chart from a new patient. An old patient underwent a surgical intervention the previous day and had a gamma nail implanted. The instructions of the doctor were clear: orthostatism the first day after the intervention. After reviewing the clinical history and radiographies, the physiotherapist did not agree with the instructions of the doctor, but he still made the patient stand up instead of talking with the doctor. I guess exposing the problem to the doctor would surely be an unnecessary waste of time” (Participant 22).
2	Condensation	Old patient with a trochanteric nail implanted the previous day. Treatment instructions established by the doctor. Based on the clinical data, the physiotherapist did not agree, but he followed the instructions without trying to reach an agreement.
3	Sub-category	Follow non-appropriate treatment instructions of a work superior
4	Category	Professional responsibility

## Data Availability

Not Applicable.

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
