# Peer review of "Physiotherapy Students’ Experiences about Ethical Situations Encountered in Clinical Practices"

_ijerph, 2021, doi:10.3390/ijerph18168489_

Round 1

Reviewer 1 Report

The paper presents the study conducted on ethical situations among physiotherapists: results have been well reelaborated and the research is interesting. In the results analysis comments on categories are very succint, but fine; in my opinion some categories reflect cultural and contextual relevant traits, as the one of sincerity, and that could be undescored and stressed by the authors, because it represents an important issue in 'ethical situations'. I suggest a revision of the words in two passages: at p. 3, line 125, the verb is 'ensure' or 'assure'? At p. 5, line 169, maybe better to change the term "the best good", that is very unfrequent for the ethical and bioethical debate. 

Author Response

REVIEWER 1:

The paper presents the study conducted on ethical situations among physiotherapists: results have been well reelaborated and the research is interesting. In the results analysis comments on categories are very succinct, but fine; in my opinion some categories reflect cultural and contextual relevant traits, as the one of sincerity, and that could be undescored and stressed by the authors, because it represents an important issue in 'ethical situations'.

Answer: We thank the reviewer´s suggestion and we absolutely agree that “sincerity”, as the rest of the categories in which ethical situations have been classified, reflects cultural and contextual relevant traits. Then, to emphasize “sincerity” we have added the following introductory sentence:

“The last category, but not the least important because of its cultural and contextual relevant traits, was sincerity” (line 257-258).

I suggest a revision of the words in two passages: at p. 3, line 125, the verb is 'ensure' or 'assure'?

Answer: We appreciate the comment. The correct verb is “ensure”. We have deleted the incorrect verb (assure) from the manuscript. Then, the following sentence has been included in the manuscript:

“In order to ensure the principle of…” (line 129)

At p. 5, line 169, maybe better to change the term "the best good", that is very unfrequent for the ethical and bioethical debate.

Answer: We thank the reviewer´s suggestion. We understand that “the best good” is a controversial concept whose difficulty often lies in what “good” means for the patient. In our context, we refer to do the best for each individual patient. Therefore, we have changed the sentence as follows:

“Students warned that the physiotherapist was not always trying to do the best for each patient (principle of beneficence) (line 173-174).

Reviewer 2 Report

I rate the article sent me for review very highly. It raises an extremely important topic. This type of research is important both from a scientific and practical point of view. I literally have a few minor comments.

  1. In the section "2.1. Participants" a more comprehensive description of the sample would be useful. it is mainly about demographic data such as age, place of residence, etc.
  2. 2. In the "2.2. Design" section, I propose to consider supplementing with a few sentences the description of why it was decided to conduct qualitative and not quantitative research
  3. 3. In the section "4.2. Strengths and limitations" I propose to add a paragraph about future research perspectives. Perhaps it is worth proposing a project based on quantitative research

Author Response

REVIEWER 2:

I rate the article sent me for review very highly. It raises an extremely important topic. This type of research is important both from a scientific and practical point of view. I literally have a few minor comments:

  1. In the section "2.1. Participants" a more comprehensive description of the sample would be useful. it is mainly about demographic data such as age, place of residence, etc.

Answer: We agree with the reviewer’s suggestion and, in order to give a more comprehensive description of the sample within the “2.1 Participants” section, we have specified that all the third-course Physiotherapy students included were students from the University of Valencia (Spain). Regarding students’ age and gender, we want to underline that in the “3. Results” section this information is described: “(mean age 23,34 ± 4,20 years, 59% women)”. Then, we have added “of Valencia (Spain)” in the following sentence:

“…were to be studying the Physiotherapy Degree at the University of Valencia (Spain)” (line 83)”

  1. In the "2.2. Design" section, I propose to consider supplementing with a few sentences the description of why it was decided to conduct qualitative and not quantitative research.

Answer: Following the reviewer´s advice we have clarified this issue into the “Design section”, by adding these sentences (lines 88-92):

“An in-depth qualitative exploratory and descriptive study was performed in order to explore ethical issues experienced by physiotherapy students. In this regard, the research was addressed to analyze students’ feelings and perceptions to identify and understand the meanings that are attributed to the focus of the study, i.e., the ethical situations encountered by the students within the clinical setting.”

  1. In the section "4.2. Strengths and limitations" I propose to add a paragraph about future research perspectives. Perhaps it is worth proposing a project based on quantitative research.

Answer: Following the reviewer’s advice, we have added the following paragraph:

“Future research about differences between ethical situations perceived by students of different academic years could be necessary to find out if their experience in the clinical setting helps them to be more or less critical from an ethical point of view. Quantitative research about the association existing between ethical situations encountered by physiotherapy students and level of knowledge in ethics could be implemented. Future studies could use mixed methods, comparing results for quantitative and qualitative content” (line 338-343).
